# From Verifiable Dot to Reward Chain: Harnessing Verifiable Reference-based Rewards for Reinforcement Learning of Open-ended Generation

**Yuxin Jiang**[1], **Yufei Wang**[1], **Qiyuan Zhang**[2], **Xingshan Zeng**[1], **Liangyou Li**[1],
**Jierun Chen**[1], **Chaofan Tao**[1], **Haoli Bai**[1], **Lifeng Shang**[1]
[1]Huawei Technologies Co.,Ltd, [2]City University of Hong Kong
{jiang.yuxin2, yufei1}@huawei.com, qzhang732-c@my.cityu.edu.hk

## Abstract

Reinforcement learning with verifiable rewards (RLVR) succeeds in reasoning tasks (e.g., math and code) by checking the final verifiable answer (i.e., a verifiable *dot* signal). However, extending this paradigm to open-ended generation is challenging because there is no unambiguous ground truth. Relying on single-dot supervision often leads to inefficiency and reward hacking. To address these issues, we propose reinforcement learning with verifiable reference-based rewards (**RLVRR**). Instead of checking the final answer, RLVRR extracts an ordered linguistic signal from high-quality references (i.e, reward chain). Specifically, RLVRR decomposes rewards into two dimensions: *content*, which preserves deterministic core concepts (e.g., keywords), and *style*, which evaluates adherence to stylistic properties through LLM-based verification. In this way, RLVRR combines the exploratory strength of RL with the efficiency and reliability of supervised fine-tuning (SFT). Extensive experiments on more than 10 benchmarks with Qwen and Llama models confirm the advantages of our approach. RLVRR (1) substantially outperforms SFT trained with ten times more data and advanced reward models, (2) unifies the training of structured reasoning and open-ended generation, and (3) generalizes more effectively while preserving output diversity. These results establish RLVRR as a principled and efficient path toward verifiable reinforcement learning for general-purpose LLM alignment. We release our code and data at https://github.com/YJiangcm/RLVRR.

## 1 Introduction

Reinforcement learning with verifiable rewards (RLVR) (Shao et al., 2024; Yu et al., 2025a; Team, 2025) has emerged as a promising paradigm for enhancing large language models (LLMs) in reasoning tasks such as mathematics and code generation. At its core, RLVR sidesteps the complicated Chain-of-Thought (CoT) supervision and only checks the correctness of the final reasoning result (i.e., the verifiable *dot*) within the reasoning solution. The presence of unambiguous ground truth makes such a verifiable dot a reliable signal, guiding exploration toward correct CoTs while preventing drift into spurious reasoning paths.

While RLVR is simple yet effective for reasoning tasks (e.g., math and code generation), it fails in open-ended generation tasks, where no unambiguous ground truth exists and reliable verification cannot be reduced to a single dot. In many cases, high-quality responses in open-ended generation should satisfy a list of content requirements simultaneously; for instance, a safe-response policy answer should explain the risk, refuse the harmful request, cite the relevant rule, and offer a safer alternative. In practice, researchers often resort to reinforcement learning from human feedback (RLHF) (Christiano et al., 2017; Bai et al., 2022; Ouyang et al., 2022) using preference-based reward models (Liu et al., 2024; 2025a) or generative reward models (Jia et al., 2025; Gunjal et al., 2025). Despite their widespread adoption, reward models suffer from two major limitations: (1) they are prone to reward hacking, often overfitting superficial artifacts and spurious correlations (Chen et al.,

2024); (2) they require large-scale pairwise annotations, making training costly and brittle during RL optimization. This motivates a critical research question: *how can we extend RL optimization to open-ended generation by moving beyond single-dot supervision?*

To this end, we introduce **RLVRR** (reinforcement learning with verifiable reference-based rewards), a framework that extends RLVR to open-ended generation. Instead of relying on a single verifiable *dot*, RLVRR extracts an ordered sequence of verifiable linguistic signals from high-quality references, transforming the *dot* supervision into a *reward chain*, akin to how mathematical reasoning derives rules from ground truth. A *reference* is a high-quality exemplar for the same prompt, which can be drawn from synthetic instruction-following corpora (e.g., OpenHermes, Magpie, WebR) (Teknium, 2023; Xu et al., 2025; Jiang et al., 2025) at scale and low cost. Mechanistically, RLVRR mirrors the single-dot principle: the reward chain anchors exploration to a standardized, verifiable checklist derived from the reference. To make supervision both reliable and efficient, RLVRR decomposes rewards into two complementary dimensions: content and style. The **content** reward uses reference-derived key points (e.g., key entities or keywords) to score a rollout by whether those deterministic core concepts are present, which remains flexibility in phrasing and expression; The **style** reward runs a small set of LLM-generated, verifiable Python checks on the rollout to confirm adherence to reference-specific stylistic properties (e.g., length, format). By integrating these complementary signals, RLVRR retains RL's exploratory dynamics but injects SFT-like token-level guidance, yielding lightweight reward, stable learning, and better generalization.

Comprehensive experiments across over 10 benchmarks show that: (1) RLVRR substantially outperforms SFT with $10\times$ more data, advanced reward models, and confidence-based rewards; (2) RLVRR can be effectively integrated into RLVR, unifying the training of structured reasoning and open-ended generation; (3) RLVRR eliminates loading reward models during RL training, incurring merely 0.71% computational overhead compared to random rewards. Moreover, our in-depth analyses reveal why RLVRR generalizes more effectively and confirm that it preserves output diversity despite relying on rule-based verifiers, underscoring its practical potential.

## 2 RELATED WORK

**Reinforcement learning with verifiable rewards.** RLVR has demonstrated strong capabilities on reasoning tasks such as math and code (Shao et al., 2024; Yu et al., 2025a; Team, 2025). By leveraging deterministic verifiers like Math-Verify (Kydlíček, 2024) and SandboxFusion (Bytedance-Seed-Foundation-Code-Team et al., 2025), RLVR enables direct correctness evaluation. Building on this paradigm, recent work has extended RLVR to broader reasoning domains. For instance, (Su et al., 2025; Ma et al., 2025) train specialized LLMs as verifier models to assess whether generated responses are equivalent to reference answers. VeriFree (Zhou et al., 2025) and RLPR (Yu et al., 2025b) bypass answer verification by leveraging policy likelihood for reference answer as a reward signal. However, these methods merely conduct experiments on datasets comprising short-form answers (nearly 10 words), overlooking the challenges of open-ended generation.

**Reinforcement learning for open-ended generation.** A pivotal advancement in applying reinforcement learning (RL) to open-ended generation is RLHF, which leverages human preference data to train a reward model that guides policy optimization. While effective, RLHF introduces several drawbacks including high training costs and susceptibility to reward hacking (Gao et al., 2023). These challenges have spurred the development of offline methods such as Direct Preference Optimization (DPO) (Rafailov et al., 2023), which optimizes policies directly from preference data without an explicit reward model. More recently, (Chang et al., 2025) directly uses BLEU (Papineni et al., 2002) between the reference and the rollout as a reward signal for open-ended tasks. Despite its simplicity, $n$-gram precision metrics such as BLEU fail to capture key content aligned with human preferences, resulting in misaligned and noisy reward signals during training.

## 3 METHODOLOGY

We propose reinforcement learning with verifiable reference-based rewards (RLVRR), a framework designed to provide reliable, low-cost rewards for open-ended generation by leveraging **Reward Chain** extracted from reference responses. RLVRR decomposes the reward signal into **content** and

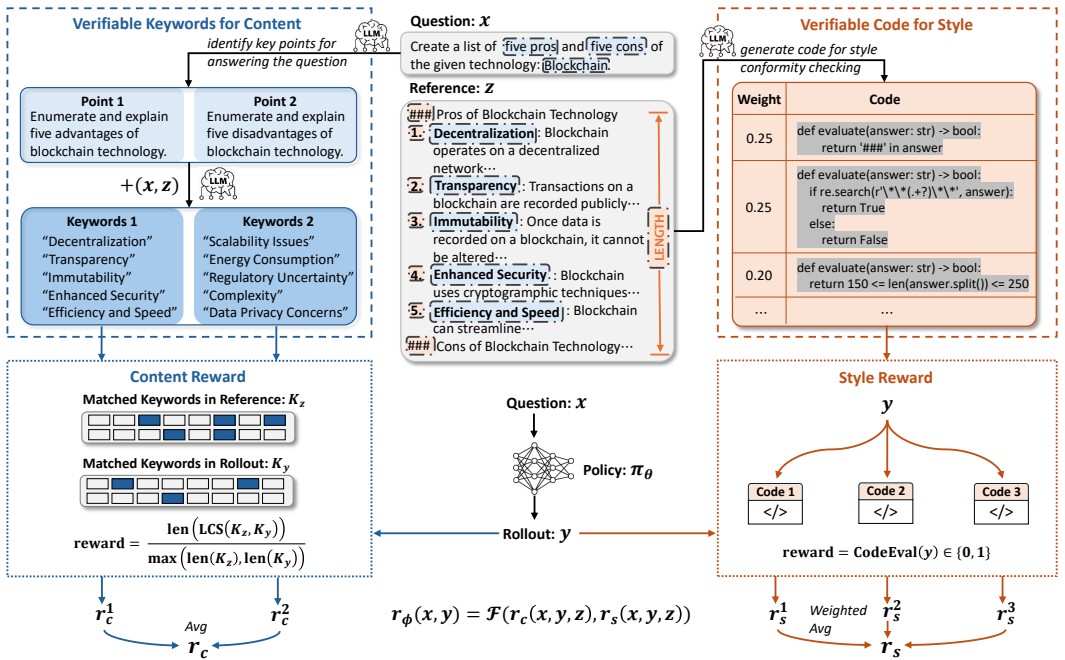

Figure 1: Overview of our proposed RLVRR framework. (1) **Upper (data construction)**: given a Question $x$ and a Reference $z$, we use an off-the-shelf LLM to generate verifiable components in terms of content and style for open-ended generation. (2) **Lower (RL training)**: these verifiable components are leveraged to calculate the rule-based reward of the Rollout $y$.

**style** dimensions, each computed through rule-based verification rather than subjective model-based scoring, as illustrated in Figure 1. We first introduce the problem formalization of RLVRR in §3.1, followed by illustrating the details of content and style rewards in §3.2 and §3.3.

## 3.1 PROBLEM FORMULATION

Let $x$ denote an open-ended instruction sampled from the training data $\mathcal{D}$. Our goal is to train a policy $\pi_\theta$ that generates a response $y$ maximizing the RL objective:

$$\mathcal{J}(\theta) = \mathbb{E}_{x \sim \mathcal{D}, y \sim \pi_\theta(\cdot|x)}[r_\phi(x, y)] - \beta \mathbb{D}_{\mathrm{KL}}\left[\pi_\theta(y \mid x) \,\|\, \pi_{\mathrm{ref}}(y \mid x)\right], \tag{1}$$

where $r_\phi$ is the reward function, $\mathbb{D}_{\mathrm{KL}}$ is the KL divergence, and $\pi_{\mathrm{ref}}$ denotes the reference model. Unlike conventional RLHF methods that rely on learned reward models to instantiate $r_\phi$, RLVRR derives rewards directly from **verifiable linguistic signals** based on a reference answer $z$:

$$r_\phi(x, y) = \mathcal{F}\left(r_c(x, y, z), r_s(x, y, z)\right), \tag{2}$$

where $r_c$ and $r_s$ quantify content fidelity and stylistic conformity, respectively, and $\mathcal{F}$ denotes the aggregation function (simple averaging in our experiments). Since both $r_c$ and $r_s$ are computed via reference-grounded rule-based reward, **RLVRR greatly mitigates the reward hacking and the inefficiency of reward models**, enabling robust and scalable RL training.

## 3.2 CONTENT REWARD OF RLVRR

**Verifiable keywords for content.** Numerous studies have shown that active learning, engaging with core concepts and rephrasing information, is more effective than passive memorization (Miller, 1956; Newport, 1990). Building on this insight, we propose a novel approach to verifiable reward design: our method extracts critical keywords (or phrases) from reference responses and optimizes the policy to maximize their inclusion during reinforcement learning. Rather than directly selecting keywords that loosely capture the semantics of the reference, we propose a novel **two-level hierarchical extraction** method: (1) an LLM first identifies a set of essential *key points* $\{p^m\}_{m=1}^M$ that

the AI assistant must address when answering the question (See prompt in Figure 4); (2) for each key point $p^m$, the LLM extracts a set of *keywords* $K^m$ (each fewer than three words) from the reference answer that encode the core facts, concepts, or entities required to assess the correctness and relevance of the response (See prompt in Figure 5). This strategy enables broader and more systematic keyword coverage while decomposing the content reward into fine-grained, verifiable units. As shown in Table 3, this separation significantly improves performance by 0.9 points. On average, the extracted keywords constitute approximately **15%** of the reference response, striking a balance between coverage and conciseness.

**Content reward calculation.**    To assess content fidelity during RL training, we propose a reward function based on keyword alignment between a generated rollout $y$ and reference text(s) $z$. For each key point $p^m$ (where $m \in [1, M]$), we extract the matched keyword sequences $K_y^m$ from $y$ and $K_z^m$ from $z$ using regular expression matching. Crucially, $K_y^m$ and $K_z^m$ **preserve both the frequency and sequential order** of matched keywords, ensuring fine-grained alignment evaluation. For each key point $p^m$, we compute the semantic coherence between $y$ and $z$ using the longest common subsequence (LCS) metric (Wagner & Fischer, 1974). LCS is chosen because it inherently captures keyword ordering and repetition, making it well-suited for evaluating the structural and semantic fidelity of generated text. The alignment score for $p^m$ is given by the normalized LCS length, while the overall content reward $r_c(x, y, z)$ is defined as the mean alignment score across all key points:

$$r_c(x, y, z) = \frac{1}{M} \sum_{m=1}^{M} \frac{\text{len}\left(\text{LCS}\left(K_z^m, K_y^m\right)\right)}{\max\left(\text{len}\left(K_z^m\right), \text{len}(K_y^m)\right)}. \tag{3}$$

**To improve robustness and accommodate multiple references** $\{z_i\}_{i=1}^{I}$, we extend Eq. (3) by selecting the highest alignment score per key point across all references. This ensures tolerance to variations in reference phrasing while maintaining rigorous content fidelity assessment:

$$r_c(x, y, \{z_i\}_{i=1}^{I}) = \frac{1}{M} \sum_{m=1}^{M} \max_i \left[ \frac{\text{len}\left(\text{LCS}\left(K_{z_i}^m, K_{y_i}^m\right)\right)}{\max\left(\text{len}\left(K_{z_i}^m\right), \text{len}\left(K_{y_i}^m\right)\right)} \right]. \tag{4}$$

In RLVRR, we set $I = 3$ and show in Table 3 that multiple references consistently improve policy performance, suggesting diversified references enhance robustness.

### 3.3    Style Reward of RLVRR

**Verifiable code for style.**    Unlike reasoning tasks, stylistic quality significantly influences model performance in open-ended generation tasks. To quantify stylistic alignment, we employ an LLM to generate a set of verifiable Python functions $\{\text{CodeEval}_n(\cdot)\}_{n=1}^{N}$, each assessing whether the rollout $y$ adheres to stylistic properties of a reference $z$. These properties include answer length, markdown formatting, and other measurable features (See prompt in Figure 6). Additionally, the LLM assigns a weight $w_n$ to each $\text{CodeEval}_n(\cdot)$, reflecting its relative importance—an approach validated empirically in our ablation study (Table 3). While our current implementation focuses on verifiable stylistic elements, semantic aspects such as tone are implicitly captured through the content reward.

**Style reward calculation.**    During reinforcement learning, we compute the style reward $r_s(x, y, z)$ by evaluating $y$ against each $\text{CodeEval}_n(\cdot)$ and aggregating the results as a weighted sum:

$$r_s(x, y, z) = \sum_{n=1}^{N} w_n \cdot \text{CodeEval}_n(y). \tag{5}$$

## 4    Experiments

### 4.1    Experimental Setup

**Models and training data.**    We conduct experiments using the Qwen2.5 (Qwen Team, 2024) and Llama3.1 (Dubey et al., 2024) model series to ensure fair comparisons with prior work and enable comprehensive evaluation. For training data, we adopt the dataset released by (Jiang et al.,

2025), comprising 100K open-ended instruction-response pairs curated from diverse high-quality instruction-tuning datasets. All responses are regenerated by GPT-4o-mini to maintain consistency in response quality. During the data construction of RLVRR, we also leverage GPT-4o-mini as the off-the-shelf LLM to generate verifiable components. Besides, we **cross-validate the quality of the verifiable components using the reference**, filtering out cases where both content and style rewards of the reference fall below 0.7. Finally, we randomly sample 10K data for RL training, where GRPO (Shao et al., 2024) is applied as the optimization algorithm to ensure that all other settings are consistent with our approach.

**Evaluation benchmarks.** We assess our models using five of the most popular open-ended instruction-following benchmarks: AlpacaEval 2 (Li et al., 2023), Arena-Hard (Li et al., 2024), MT-Bench (Zheng et al., 2023), IFEval (Zhou et al., 2023), and FollowBench (Jiang et al., 2024). For AlpacaEval 2, we report the length-controlled win rate (LC), which ensures robustness against verbosity. For Arena-Hard, we report the win rate (WR) against the baseline model. For MT-Bench, we provide the average score, using GPT-4.1-mini as the evaluation judge. For IFEval and FollowBench, we report the prompt-level strict accuracy and the hard satisfaction rate, respectively. Besides, we evaluate the impact of diverse methods on tasks across multiple domains: (1) **Knowledge**: MMLU (Hendrycks et al., 2021a); (2) **Reasoning**: ARC (Clark et al., 2018); (3) **Math**: MATH (Hendrycks et al., 2021b); (4) **Code**: HumanEval (Chen et al., 2021). More evaluation details are listed in Appendix B.

**Baselines.** We compare RLVRR with seven established and contemporaneous methods, categorized into SFT, reward strategies, and DPO. (1) **SFT**: Standard supervised fine-tuning (Wei et al., 2022; Mishra et al., 2022) on (i) 10K data which shares identical prompts with RL, or (ii) 100K data. (2) **Random**: We examine whether random rewards $\sim$ Uniform$(0, 1)$ can benefit open-ended generation. (3) **BLEU**: (Chang et al., 2025) directly uses BLEU (Papineni et al., 2002) between the reference and the rollout as a reward signal for RL-based alignment. (4) **RM**: We use Skywork-Reward-V2-Llama-3.1-8B[1] (Liu et al., 2025a) trained on well-curated preference data as the reward model to score output in GRPO. (5) **GRM**: Following Rubrics as Rewards (Gunjal et al., 2025), we use GPT-4o-mini as the generative reward model to judge whether the rollout satisfies checklist-style rubrics. (6) **RLPR** (Yu et al., 2025b): RLPR is a verifier-free framework that uses the LLM's own token probability scores of reference answers as the reward signal. (7) **DPO** (Rafailov et al., 2023): We generate the preference dataset following (Meng et al., 2024). For each question $x$, we first generate 5 responses using the INSTRUCT model and then use GPT-4o-mini to select the best one as *win* and the worst one as *lose*. All implementation details are illustrated in Appendix A.1.

## 4.2 MAIN RESULTS

Table 1 summarizes the performance of various methods across five open-ended benchmarks and four additional tasks, revealing several key findings. (1) **Superiority over SFT**: Remarkably, RLVRR outperforms SFT by a significant margin on open-ended tasks, even when SFT is trained with *10× more data*. (2) **Advantages over alternative reward strategies**: RLVRR consistently surpasses other reward strategies, including random reward, BLEU, reward model (RM), generative reward model (GRM), and RLPR. Notably, it improves over the RM-based approach—which requires loading an auxiliary reward model during training—by **+2.3** and **+2.7** points on Qwen2.5-3B-Base and Instruct, respectively. (3) **Improved over DPO**: RLVRR exhibits stronger performance than DPO, a widely adopted alignment method, further validating its effectiveness. (4) **Robustness across scales and initializations**: The benefits of RLVRR persist across varying model sizes and training starting points, demonstrating its general applicability. (5) **Generalization to diverse tasks**: Beyond open-ended generation, RLVRR achieves state-of-the-art results on knowledge-intensive, reasoning, mathematical, and coding tasks, highlighting its superior generalization capability.

## 4.3 INTEGRATION WITH MATHEMATICAL REASONING

To examine the compatibility of our method in jointly optimizing for both closed-form reasoning and open-ended generation within RLVR, we focus on the mathematical domain as a representative setting. The reasoning template is shown in Appendix A.3. Following SimpleRL-Zoo (Zeng et al.,

---

[1]This model ranks first on RewardBench (Lambert et al., 2025) as of September 24th, 2025.

Table 1: Evaluation results across five open-ended benchmarks and four other tasks. The results of Llama3.1, which indicate consistent findings, are shown in Appendix C.

| Method | #Data | Alpaca Eval 2 | Arena Hard | MT Bench | IF Eval | Follow Bench | Avg. | MMLU | ARC | MATH | Human Eval | Avg. |
|---|---|---|---|---|---|---|---|---|---|---|---|---|
| **Qwen2.5-3B Models** | | | | | | | | | | | | |
| *Base* | - | 0.8 | 6.5 | 6.4 | 22.0 | 12.4 | 9.6 | 66.8 | 75.6 | 54.0 | **66.5** | 65.7 |
| ↪ SFT | 10K | 22.0 | 27.3 | 7.5 | 31.8 | 45.0 | 26.7 | 66.1 | 83.7 | 59.6 | 65.9 | 68.8 |
| ↪ SFT | 100K | **25.1** | 32.9 | 7.5 | 35.9 | **51.3** | 30.5 | 60.4 | 81.4 | 58.7 | 65.9 | 66.6 |
| ↪ GRPO (Random) | 10K | 3.7 | 3.6 | 6.1 | 25.7 | 16.1 | 11.0 | 66.9 | 73.7 | 59.7 | 61.6 | 65.5 |
| ↪ GRPO (BLEU) | 10K | 14.4 | 26.6 | 6.9 | 29.2 | 41.8 | 23.8 | 67.2 | 82.0 | 59.9 | 62.8 | 68.0 |
| ↪ GRPO (RM) | 10K | 22.4 | 33.7 | 7.3 | 32.8 | 47.6 | 28.8 | 67.1 | 84.2 | 59.6 | 65.1 | 69.0 |
| ↪ GRPO (GRM) | 10K | 21.1 | 30.5 | 7.4 | 35.4 | 47.3 | 28.3 | 65.5 | 81.2 | 58.2 | 63.9 | 67.2 |
| ↪ GRPO (RLPR) | 10K | 21.8 | 28.6 | 7.4 | 32.6 | 47.2 | 27.5 | 65.7 | 82.8 | 58.7 | 65.3 | 68.1 |
| ↪ GRPO (RLVRR) | 10K | 23.7 | **35.3** | **7.6** | 37.7 | 51.2 | **31.1** | 67.9 | 85.7 | 60.6 | 66.0 | **70.0** |
| *Instruct* | - | 17.0 | 19.3 | 7.8 | 54.9 | 47.5 | 29.3 | 67.3 | 84.8 | 63.2 | 71.3 | 71.6 |
| ↪ DPO | 10K | 18.0 | 31.1 | 7.6 | 59.3 | 49.6 | 33.1 | 67.1 | 84.8 | **63.7** | 69.2 | 71.2 |
| ↪ GRPO (RM) | 10K | 22.3 | 34.1 | 7.6 | 55.3 | 49.3 | 33.7 | 67.5 | 85.3 | 63.2 | 70.7 | 71.7 |
| ↪ GRPO (RLVRR) | 10K | **24.3** | **36.5** | **7.9** | **61.3** | **51.9** | **36.4** | 67.8 | 85.4 | 63.6 | **71.9** | **72.2** |
| **Qwen2.5-7B Models** | | | | | | | | | | | | |
| *Base* | - | 2.1 | 8.9 | 7.3 | 24.7 | 14.9 | 11.6 | 74.2 | 79.8 | 69.4 | 76.0 | 74.9 |
| ↪ SFT | 10K | 30.0 | 53.2 | 8.3 | 42.3 | 47.5 | 36.3 | 75.2 | **89.8** | 67.5 | 77.4 | 77.5 |
| ↪ SFT | 100K | 32.3 | 52.0 | 8.3 | 43.5 | **56.3** | 38.5 | 70.9 | 87.5 | 67.6 | 76.7 | 75.7 |
| ↪ GRPO (Random) | 10K | 4.5 | 7.8 | 7.4 | 28.3 | 15.0 | 12.6 | 74.3 | 78.6 | 68.3 | 76.6 | 74.4 |
| ↪ GRPO (BLEU) | 10K | 19.9 | 44.1 | 7.8 | 39.5 | 46.8 | 31.6 | 74.6 | 83.5 | 68.0 | 77.1 | 75.8 |
| ↪ GRPO (RM) | 10K | 32.8 | 53.5 | 8.2 | 43.2 | 49.0 | 37.3 | 74.8 | 88.3 | 68.8 | 76.5 | 77.1 |
| ↪ GRPO (GRM) | 10K | 31.6 | 52.7 | 8.1 | 43.9 | 49.5 | 37.2 | 73.6 | 86.4 | 68.8 | 76.2 | 76.3 |
| ↪ GRPO (RLPR) | 10K | 31.9 | 51.7 | 8.2 | 42.6 | 49.2 | 36.7 | 72.4 | 87.0 | 67.1 | 76.1 | 75.7 |
| ↪ GRPO (RLVRR) | 10K | **33.6** | **54.9** | **8.3** | 47.8 | 54.6 | **39.8** | 75.7 | 89.6 | 70.1 | 77.5 | **78.2** |
| *Instruct* | - | 35.6 | 37.1 | 8.7 | 69.7 | 53.8 | 41.0 | 74.9 | 90.2 | 80.6 | 83.8 | 82.4 |
| ↪ DPO | 10K | 36.7 | 52.4 | 8.2 | 69.3 | 53.3 | 44.0 | 74.3 | 89.9 | **80.9** | 82.6 | 81.9 |
| ↪ GRPO (RM) | 10K | 37.6 | 53.6 | 8.4 | 69.1 | 53.9 | 44.5 | 75.1 | 89.5 | 80.2 | 82.8 | 81.9 |
| ↪ GRPO (RLVRR) | 10K | **41.4** | **55.8** | **8.8** | **70.3** | **55.7** | **46.4** | 75.6 | 90.3 | 80.6 | **84.1** | **82.6** |

Table 2: Performance comparison of math tasks based on Qwen2.5-3B-Base.

| Method | GSM8K | MATH 500 | Minerva Math | GaoKao 2023 En | Olympiad Bench | College Math | Avg. | Open-ended Avg. |
|---|---|---|---|---|---|---|---|---|
| *Base* | 81.8 | 61.2 | 21.0 | 48.1 | 25.0 | 39.5 | 46.1 | 9.6 |
| ↪ GRPO - 10K math (RLVR) | 86.2 | **68.0** | 26.1 | **57.4** | **30.7** | 43.2 | 51.9 | 22.6 |
| ↪ GRPO - 10K math (RLVRR) | 86.0 | 66.2 | 25.8 | 55.8 | 29.6 | 43.2 | 51.1 | 25.3 |
| ↪ GRPO - 10K open-ended (RLVRR) | 85.0 | 64.0 | 25.4 | 54.3 | 26.8 | 43.0 | 49.8 | **31.1** |
| ↪ GRPO - 5k math (RLVR) + 5k open-ended (RLVRR) | 86.0 | 67.8 | **29.4** | 57.3 | 28.0 | 42.7 | **51.9** | 30.7 |
| ↪ GRPO - 5k math (RLVR) + 5k open-ended (RM) | 84.6 | 67.0 | 24.7 | 56.5 | 27.3 | 42.3 | 50.4 | 28.2 |
| *Instruct* | **87.0** | 64.8 | 27.6 | 56.6 | 27.3 | **45.1** | 51.4 | 29.3 |

2025), we stratify the MATH dataset (Lewkowycz et al., 2022) into five difficulty levels and randomly sample 10K examples from levels 2–5 as the base for math-focused RL training. To explore integration, we construct a mixed training set by combining 5k math-focused samples (*using rule-based reward*) with 5k open-ended instances (*using RLVRR-based reward*). We evaluate the resulting models on six standard mathematical reasoning benchmarks, including GSM8K (Cobbe et al., 2021), MATH 500 (Hendrycks et al., 2021b), Minerva Math (Lewkowycz et al., 2022), GaoKao 2023 En (Liao et al., 2024), Olympiad Bench (He et al., 2024), and College Math (Tang et al., 2024). We report the performance of CoT reasoning with greedy decoding.

As shown in Table 2, RLVR trained solely on mathematical data significantly boosts performance on math benchmarks but generalizes poorly to open-ended tasks (Avg. 22.6). In contrast, RLVRR trained only on open-ended data achieves strong performance in open-ended tasks and also improves mathematical reasoning (Avg. 49.8), indicating positive transfer. Unified training on mixed data provides the best balance, reaching 51.9 on math benchmarks and 30.7 on open-ended tasks. Remarkably, this setting even surpasses the INSTRUCT model trained on millions of samples, despite using only 10K RL training instances. Furthermore, RLVRR demonstrates better compatibility with reasoning tasks compared to RM. These results demonstrate that **our method seamlessly integrates with RLVR, unifying the training of structured reasoning and open-ended generation**.

Table 3: Ablation study based on Qwen2.5-3B-Base.

| Method | AlpacaEval 2 | Arena-Hard | MT-Bench | IF-Eval | FollowBench | Avg. |
|---|---|---|---|---|---|---|
| GRPO (RLVRR) | **23.7** | **35.3** | **7.6** | **37.7** | 51.2 | **31.1** |
| *Effect of content reward and style reward* | | | | | | |
| -w/o content reward | 9.6 | 10.2 | 6.7 | 28.6 | 35.5 | 18.1 |
| -w/o multiple references | 23.6 | 35.1 | 7.6 | 36.1 | 50.9 | 30.7 |
| - repl. LCS with direct matching | 10.3 | 8.5 | 6.5 | 29.0 | 34.7 | 17.8 |
| -w/o style reward | 19.6 | 28.5 | 6.6 | 36.6 | 50.1 | 28.3 |
| -w/o weight in style | 22.2 | 35.0 | 7.6 | 36.2 | 48.6 | 29.9 |
| *Effect of keyword extraction* | | | | | | |
| -w/o two-level extraction | 22.6 | 35.3 | 7.5 | 34.3 | **51.3** | 30.2 |
| - extract 15% keywords randomly | 19.0 | 32.4 | 7.1 | 32.9 | 43.7 | 27.0 |
| - extract 15% keywords by TF-IDF | 19.6 | 31.3 | 7.4 | 33.3 | 45.5 | 27.4 |
| - extract 30% keywords by TF-IDF | 19.4 | 30.5 | 7.3 | 32.7 | 45.9 | 27.2 |

# 5 ANALYSIS

## 5.1 ABLATION STUDY

To systematically evaluate method components and offer a comprehensive understanding of RLVRR, we conduct ablation studies based on Qwen2.5-3B-Base in Table 3.

**Effect of content reward.** Our ablation study reveals that removing the content reward results in a severe performance degradation, with the average score dropping by 13.0 points compared to the full method. This underscores the critical role of content alignment in response generation. Interestingly, when using only a single reference (instead of multiple references) for content reward computation, performance remains robust, declining marginally from 31.1 to 30.7, demonstrating the method's resilience to reference variability. Finally, we attempt to replace LCS with a naïve "direct matching" approach, which calculates the percentage of keywords appearing in the rollout as the content reward. This approach leads to catastrophic failure as it (1) disregards keyword ordering and (2) incentivizes reward hacking (Skalse et al., 2022), where the model generates excessively verbose outputs to artificially inflate keyword coverage.

**Effect of style reward.** The absence of style reward reduces performance by 2.8 points, confirming that learning presentation, structure, and formatting from references is essential for high-quality responses. Moreover, when style reward components are aggregated without LLM-generated importance weights, performance drops by 1.2 points, validating that LLM-derived weighting effectively captures stylistic nuances.

**Effect of keywords extraction.** We analyze the impact of keyword extraction strategies on alignment performance. First, we ablate the two-level hierarchical extraction process in favor of a single-step approach where keywords are directly extracted from the full response. This leads to a 0.9-point drop in average score, confirming that hierarchical extraction improves keyword precision and coverage. Next, we compare LLM-based extraction with rule-based alternatives: (1) **random selection** after stopword filtering and (2) **TF-IDF-based selection** (Sparck Jones, 1972), both extracting 15% of words for fairness. As shown in Table 3, LLM extraction outperforms both variants by 3.7–4.1 points, demonstrating its superiority in identifying semantically critical keywords. Notably, increasing the TF-IDF keyword ratio to 30% further degrades performance, suggesting that **quality matters more than quantity**—a sparse set of high-value keywords suffices for effective learning.

**Effect of reference LLMs.** Table 4 examines the impact of diverse reference LLMs, where the LLM is used for (1) reference generation and (2) verifiable component generation of RLVRR. Remarkably, substituting the GPT-4o-mini model with a less powerful yet open-source alternative, such as Llama3-70B-Instruct (Dubey et al., 2024), yields consistent results— **RLVRR continues to outperform SFT even when SFT is trained with 10× more data**. This demonstrates the robustness

Table 4: Impact of various reference LLMs based on Qwen2.5-3B-Base.

| Method | LLM of Ref | #Data | Open-ended Avg. |
|---|---|---|---|
| SFT | GPT-4o-mini | 10K | 26.7 |
| SFT | GPT-4o-mini | 100K | 30.5 |
| GRPO (RLVRR) | GPT-4o-mini | 10K | **31.1** |
| SFT | Llama3-70B-Inst | 10K | 24.8 |
| SFT | Llama3-70B-Inst | 100K | 28.3 |
| GRPO (RLVRR) | Llama3-70B-Inst | 10K | **28.9** |

Table 5: Results of SFT via self-data distillation based on Qwen2.5-3B-Base.

| Method | #Data | Open-ended Avg. |
|---|---|---|
| *Base* | - | 9.6 |
| ↪ SFT | 10K | 26.7 |
| ↪ SFT | 100K | **30.5** |
| ↪ SFT-distilled SFT | 10K | 25.0 |
| ↪ RLVRR-distilled SFT | 10K | 29.2 |

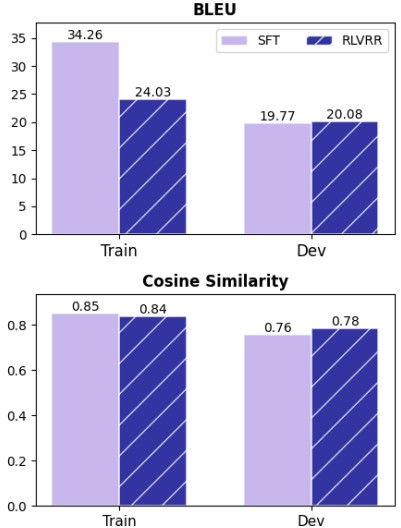

Figure 2: Average BLEU scores and semantic similarities between references and generated responses for SFT and RLVRR.

Figure 3: Curves of reward and response length during RL training with different methods.

of our approach across varying levels of LLM sophistication and highlights its potential to reduce reliance on proprietary commercial models without compromising downstream performance.

## 5.2 LEARNING WHAT MATTERS: WHY RLVRR GENERALIZES BETTER THAN SFT

In this section, we investigate why RLVRR, which reinforces quality signals (keywords or phrases), outperforms SFT, which models the entire reference sequence token-by-token. To compare their generalization behaviors, we conduct a controlled study on 1,000 randomly sampled prompts, each from the training and development sets. For each prompt, we generate responses using two models trained separately with SFT and RLVRR, and evaluate their quality with BLEU and cosine semantic similarity against references. Semantic similarity is computed using embeddings from all-mpnet-base-v2 (Reimers & Gurevych, 2019). Figure 2 show that while SFT achieves higher BLEU on the training set, this advantage vanishes and even reverses on the development set, indicating strong memorization but poor generalization (Chu et al., 2025). The limitation stems from SFT's *imitation learning* objective, which minimizes token-level prediction error under teacher forcing: $\mathcal{L}_{\text{SFT}} = -\sum_{t=1}^{|z|} \log \pi_\theta(z_t|x, z_{<t})$. This training paradigm enforces exact mimicry but suffers from *exposure bias* (Zhang et al., 2019; Schmidt, 2019), as the model never recovers from its own mistakes. RLVRR, in contrast, rewards the preservation of key semantic elements while allowing flexible phrasing, leading to stable performance across both training and development sets. Specifically, RLVRR maintains consistent BLEU and higher semantic similarity (0.84 vs. 0.85 on training; 0.78 vs. 0.76 on development, compared to SFT). These results suggest that **RLVRR better captures the semantic essence of references and generalizes more effectively to unseen inputs**.

Table 6: Average runtime per step for different reward strategies in RL training, based on Qwen2.5-3B-Base.

| Method | Time (s) | $\Delta$ Random (%) |
|---|---|---|
| Random | 121.56 | 0.00% |
| BLEU | 122.38 | 0.67% |
| RM | 131.62 | 8.28% |
| GRM | 128.92 | 6.05% |
| RLPR | 129.38 | 6.43% |
| RLVRR | 122.42 | 0.71% |

Table 7: Average best@5 performance and Self-BLEU cross five open-ended benchmarks, based on Qwen2.5-3B-Base.

| Method | Open-ended Avg. Best@5 ($\uparrow$) | Self-BLEU ($\downarrow$) |
|---|---|---|
| *Base* | 11.1 | 27.1 |
| $\hookrightarrow$ SFT | 29.8 | 24.2 |
| $\hookrightarrow$ GRPO (BLEU) | 25.2 | 26.6 |
| $\hookrightarrow$ GRPO (RM) | 30.8 | 23.9 |
| $\hookrightarrow$ GRPO (RLPR) | 29.6 | 24.5 |
| $\hookrightarrow$ GRPO (RLVRR) | **33.2** | 24.0 |
| *Instruct* | 31.0 | **23.7** |

## 5.3 SELF-DATA DISTILLED RLVRR OUTPERFORMS STANDARD SFT

Recent work such as DeepSeek-R1 (DeepSeek-AI, 2025) demonstrates that fine-tuning on trajectories sampled from the same model post-RL training, an approach we refer to as *self-data distillation*, can yield better performance than standard SFT on reasoning tasks. In this section, we extend this idea to open-ended generation and examine whether a similar benefit holds. As shown in Table 5, self-data distillation using RLVRR significantly improves performance over standard SFT (2.5-point gain in average performance) when both are trained on the same 10K dataset. While it does not match the performance of SFT trained on the full 100K data, it notably narrows the gap using only 10% of the data. These results highlight the superior quality of supervision signals produced by RLVRR. Moreover, since the distilled data remains close in distribution to the base model's outputs, the resulting student model benefits from both strong alignment and distributional consistency.

## 5.4 TRAINING DYNAMICS ANALYSIS & COST ANALYSIS

**Training dynamics analysis.** Figure 3 visualizes the curves of reward and response length during RL training with different methods (Random, BLEU, RM, and RLVRR). We observe that RLVRR achieves a more stable and substantial increase in reward compared to the other methods, highlighting its effectiveness in providing consistent and high-quality learning signals. This trend is further validated by the content/style reward curves in Figure 8. Notably, RLVRR's response length surges initially, reflecting exploratory behavior for informative outputs, then declines and stabilizes as the model learns conciseness. This demonstrates **RLVRR's robustness against reward hacking**, as it avoids exploiting length for reward gains. In contrast, RM persistently favors longer responses, likely due to over-reliance on superficial heuristics rather than true quality.

**Cost analysis.** We present a detailed cost breakdown of RLVRR in Appendix D, covering both the data construction and RL training phases. Key findings include: (1) the total cost of API calls during data construction is $21.36, which is highly economical given the scale of the task; (2) in the RL training phase, RLVRR introduces only a **0.71%** computational overhead compared to the Random Reward baseline (refer to Table 6). These results underscore RLVRR's practicality for real-world deployment, with minimal financial and computational burdens.

## 5.5 RLVRR DOES NOT COMPROMISE DIVERSITY

A potential concern with RLVRR's reference-based verifiable reward is that it could restrict output diversity. To examine this, we set the decoding temperature to 1.0 and sampled five responses per method across five open-ended benchmarks, reporting average best@5 and Self-BLEU in Table 7. The relative performance improvements of RLVRR over baselines remain consistent with Table 1. Notably, RLVRR attains a Self-BLEU of 24.0, comparable to RM (23.9) and the INSTRUCT model (23.7). These findings indicate that **RLVRR does not sacrifice diversity despite its reliance on verifiable references**, and in fact, enhances the model's ability to generate diverse responses relative to other reward strategies.

## 6 CONCLUSION

In this paper, we propose RLVRR, a novel framework that extends verifiable reward learning beyond reasoning tasks to open-ended generation. By constructing rule-based verifiers derived from high-quality references across content and style dimensions, RLVRR retains RL's exploratory dynamics but injects SFT-like token-level guidance, thus providing reliable and low-cost training signals. Our results establish RLVRR as an efficient and scalable path toward verifiable reinforcement learning for general-purpose LLMs.

## REPRODUCIBILITY STATEMENT

We are committed to ensuring the transparency and reproducibility of our research. To support this commitment, we will publicly release our annotated dataset and all source code, facilitating future extensions and community research. Comprehensive details of our methodology are provided throughout this paper: the prompts used for data construction are illustrated in Appendix A.2; the evaluation details are shown in Appendix B. Furthermore, the experimental implementations can be found in Appendix A.1. We believe that releasing these assets will lower the barrier for replication, enable fair comparisons, and foster further exploration in this line of research.

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

## APPENDICES

## A DETAILED EXPERIMENTAL SETUP

### A.1 IMPLEMENTATION DETAILS

We adopt the OpenRLHF (Hu et al., 2024) framework for efficient training. During SFT, we train models for 3 epochs with a learning rate of 2e-5, a batch size of 128, a max sequence length of 2048, and a cosine learning rate schedule with 10% warmup steps. During GRPO, we set the epoch to 1, the learning rate to 5e-7, the number of rollouts to 8, max prompt length and max generation length to 1024 tokens, and maintain the same global batch size of 128. During DPO, we train models for 1 epoch with a learning rate of 5e-7, a batch size of 128, a max sequence length of 2048, and a $\beta$ of 1e-2. All experiments are conducted on 8 NVIDIA A800 GPUs. We report the average performance of three random runs.

### A.2 PROMPT TEMPLATE FOR DATA CONSTRUCTION

---

**Prompt Template of Generating Key Points for Answering the Question**

You are given a **Question**. Your task is to identify **all essential key points** that the AI assistant must notice when answering the question. Return your output as a Python list of strings, where each string is a key point.

**[Question]:**
{question}

---

Figure 4: Prompt template of generating key points for answering the question.

---

**Prompt Template of Generating Keywords**

You are given a **Question**, **Key Points of the Question**, and a **Reference Answer**. Your task is to identify **all essential keywords (less than 3 words for each keyword)** from the reference answer that match each key point. These keywords should be explicitly mentioned or accurately reflected in a good AI-generated answer. These keywords should represent the core facts, concepts, or entities required to assess the correctness and relevance of the response.

### Output Format:
Please return your schema in the following JSON format:
json
{{
  "key_points": [
    {{
      "point": "<Key point>",,
      "keywords": <a Python list of strings>,
    }}
  ],
}}

**[Question]:**
{question}

**[Key Points]:**
{keypoint}

**[Reference Answer]:**
{reference}

---

Figure 5: Prompt template of generating keywords.

---

**Prompt Template of Generating Code for Style Conformity Checking**

You are given a **Reference Answer**. Your task is to evaluate whether an **AI-generated answer** follows a similar **format and style** as the reference answer. The goal is not to assess content correctness or completeness, but to compare the presentation, structure, and formatting features of the two answers.

### Key Evaluation Focus:
Your evaluation should focus on **style-related aspects** such as:
- Overall structure and organization (e.g., use of sections or bullet points)
- Length similarity (in terms of word count, within a reasonable range)
- Paragraph count and distribution (roughly comparable, not necessarily identical)
- Use of markdown elements like bold text, headers, lists, code blocks
- Visual layout and clarity

**Do NOT evaluate the factual content, relevance, or correctness of the answer.**

### Instructions:
1. **Identify 3–6 Key Style Evaluation Points:** For each point, define a **specific and measurable** style-related criterion. Allow for small variations instead of requiring exact matches.
2. **Define a Python Function for Each Point:** The function should be named `evaluate(answer: str) -> bool` and return:
   - `True` if the AI-generated answer satisfies the style point (even approximately),
   - `False` otherwise.
3. **Assign a Weight to Each Point:** Each point should be assigned a weight that reflects its relative importance. All weights must sum to **1.0**.

### Output Format:
Please return your evaluation schema in the following JSON format:
```json
{{
  "key_points": [
    {{
      "point": "<Key evaluation point>",
      "explanation": "<Why this point helps assess format/style similarity>",
      "verification_code": "def evaluate(answer: str) -> bool:\\n    # Your logic here\\n    return ...",
      "weight": <float weight>
    }}
  ],
  "total_weight": 1.0
}}
```

**[Reference Answer]:**
{reference}

Figure 6: Prompt template of generating code for style conformity checking.

## A.3 TEMPLATE FOR MATHEMATICAL REASONING

Figure 7 shows the training and evaluation template for mathematical reasoning, where we first require the model to think step by step and then output the final answer within "boxed{}".

---

**Training and Evaluation Template for Mathematical Reasoning**

```
<|im_start|>user
{question}
Please reason step by step, and put your final answer within \\boxed{}.<|im_end|>
<|im_start|>assistant
```

Figure 7: Training and evaluation template for mathematical reasoning.

## B    Evaluation Details

Table 8 lists the evaluation details for AlpacaEval 2 (Li et al., 2023), Arena-Hard (Li et al., 2024), MT-Bench (Zheng et al., 2023), IFEval (Zhou et al., 2023), and FollowBench (Jiang et al., 2024). AlpacaEval 2 comprises 805 questions from 5 datasets, and MT-Bench spans 8 categories with a total of 80 questions. Arena-Hard is an enhanced version of MT-Bench, featuring 500 well-defined technical problem-solving queries. IFEval comprises 541 samples designed to evaluate instruction-following LLMs through diverse, verifiable instructions that include numerous lexical and formatting constraints. FollowBench is a multi-level, fine-grained benchmark for evaluating constraint-following capabilities, featuring 820 samples across five constraint types and five difficulty levels. To balance cost and performance, we select GPT-4.1-mini as the judge. Evaluation metrics are reported in accordance with each benchmark's protocol. For tasks across multiple domains, we align our evaluation settings with OpenCompass (Contributors, 2023).

Table 8: Evaluation details for AlpacaEval 2, Arena-Hard, MT-Bench, IFEval, and FollowBench. The baseline model refers to the model compared against.

| Benchmark | # Exs. | Baseline Model | Judge Model | Scoring Type | Metric |
|---|---|---|---|---|---|
| AlpacaEval 2 | 805 | GPT-4 Turbo | GPT-4.1-mini | Pairwise comparison | Length-controlled win rate |
| Arena-Hard | 500 | GPT-4-0314 | GPT-4.1-mini | Pairwise comparison | Win rate |
| MT-Bench | 80 | - | GPT-4.1-mini | Single-answer grading | Rating of 1-10 |
| IFEval | 541 | - | - | Rule-based verification | Accuracy |
| FollowBench | 820 | - | GPT-4.1-mini | Rule and LLM verification | Satisfaction rate |

## C    Experimental Results of Llama3.1

Table 9 presents results on Llama3.1-8B-Instruct, as prior work shows that effective GRPO training requires a sufficiently strong base model (Liu et al., 2025b). RLVRR consistently outperforms all baselines by more than 2 points, with a comparable improvement observed on Qwen2.5. These findings confirm that **our approach generalizes robustly across different model architectures**.

Table 9: Evaluation results of Llama3.1-8B across five open-ended benchmarks and four other tasks.

| Method | #Data | Alpaca Eval 2 | Arena Hard | MT Bench | IF Eval | Follow Bench | Avg. | MMLU | ARC | MATH | Human Eval | Avg. |
|---|---|---|---|---|---|---|---|---|---|---|---|---|
| *Instruct* | - | 30.9 | 34.3 | 8.4 | 76.8 | 54.2 | 40.9 | 69.4 | 83.4 | 51.9 | 72.6 | 69.3 |
| ↪ SFT | 10K | 31.2 | 46.9 | 8.5 | 75.6 | 53.2 | 43.1 | 67.7 | 83.7 | 50.6 | 69.9 | 68.0 |
| ↪ SFT | 100K | 33.1 | 51.0 | 8.5 | 75.9 | 55.3 | 44.8 | 65.4 | 81.6 | 51.7 | 70.8 | 67.4 |
| ↪ GRPO (Random) | 10K | 5.2 | 6.7 | 7.6 | 26.3 | 17.5 | 12.7 | 66.7 | 79.7 | 40.7 | 66.8 | 63.5 |
| ↪ GRPO (BLEU) | 10K | 30.4 | 39.6 | 8.2 | 69.2 | 49.8 | 39.4 | 69.2 | 82.0 | 50.2 | 72.8 | 68.6 |
| ↪ GRPO (RM) | 10K | 32.5 | 50.7 | 8.7 | 76.0 | 53.7 | 44.3 | 69.6 | 84.2 | 52.4 | 72.1 | 69.6 |
| ↪ GRPO (GRM) | 10K | 31.1 | 48.5 | 8.4 | 75.4 | 54.3 | 43.5 | 69.5 | 83.2 | 51.2 | 70.9 | 68.7 |
| ↪ GRPO (RLPR) | 10K | 30.8 | 48.6 | 8.3 | 74.6 | 54.4 | 43.3 | 68.7 | 82.8 | **52.7** | 72.3 | 69.1 |
| ↪ DPO | 10K | 32.0 | 48.1 | 8.6 | 77.3 | 53.6 | 43.9 | 69.1 | 83.8 | 51.0 | 72.2 | 69.0 |
| ↪ GRPO (RLVRR) | 10K | **36.7** | **52.3** | **8.7** | **77.7** | **56.2** | **46.3** | **70.2** | **84.9** | 52.6 | **73.0** | **70.2** |

## D    Cost Analysis

### D.1    Cost of Data Construction

The data construction phase, responsible for synthesizing verifiable components for content and style reward, operates exclusively **offline**, meaning it incurs no runtime cost during model training. For context, we estimated the budget for data synthesis using the GPT-4o-mini API, based on the API's pricing of $0.15 per 1M input tokens and $0.60 per 1M output tokens. Table 10 lists the breakdown of the estimated costs, which demonstrates that the overall expenditure (**$21.36**) is both reasonable and manageable.

**Can an open-source LLM be utilized as an alternative?**    In Table 4, we explore the impact of LLMs on generating verifiable components during the data construction phase. Our findings indicate that substituting the GPT-4o-mini model with a less powerful yet open-source alternative,

Table 10: Estimated budget for data construction using the GPT-4o-mini API.

| Task | # of Samples | Avg. Input Token Length | Avg. Output Token Length | Cost ($) |
|---|---|---|---|---|
| Multiple References | 20,000 | 170 | 652 | 8.33 |
| Key Points | 10,000 | 202 | 234 | 1.71 |
| Keywords | 30,000 | 1,156 | 103 | 7.06 |
| Style | 10,000 | 853 | 497 | 4.26 |

such as Llama3-70B-Instruct, **yields comparable performance while significantly surpassing SFT trained with 10× more data**. The Llama3-70B-Instruct model can be deployed on only 2 NVIDIA 3090 GPUs, with the option to further reduce hardware requirements through low-bit quantization[2]. This provides an economical alternative for RLVRR without compromising performance. Overall, our framework demonstrates robustness in leveraging diverse LLMs for verifiable component generation, confirming its adaptability and effectiveness.

## D.2 COST OF RL TRAINING

**RLVRR incurs negligible computational overhead.** As shown in Table 11, we report the average runtime per training step on 8 NVIDIA A800 GPUs across various reward strategies. RLVRR increases runtime by only **0.71%** compared to the Random Reward baseline, comparable to the lightweight BLEU-based reward (+0.67%). In contrast, RM introduces a substantial 8.28% overhead due to the need to maintain and query a learned reward model, while RLPR incurs a 6.43% increase from additional reference forward passes. These results highlight that RLVRR achieves verifiability with minimal runtime cost, making it a scalable choice for real-world RL training scenarios.

Table 11: Average runtime per step for different reward strategies in RL training.

| Method | Time (s) | Δ Random (%) |
|---|---|---|
| Random | 121.56 | 0.00% |
| BLEU | 122.38 | 0.67% |
| RM | 131.62 | 8.28% |
| GRM | 128.92 | 6.05% |
| RLPR | 129.38 | 6.43% |
| RLVRR | 122.42 | 0.71% |

## E REWARD CURVES

Figure 8 presents the training dynamics of RLVRR in terms of **content** and **style** rewards. Both rewards exhibit a consistent upward trend in the early stages, indicating effective optimization across dimensions. Notably, the style reward plateaus after approximately 60 steps, suggesting that stylistic improvements saturate relatively quickly. In contrast, the content reward continues to increase, albeit more gradually, highlighting the model's sustained ability to refine content quality over time.

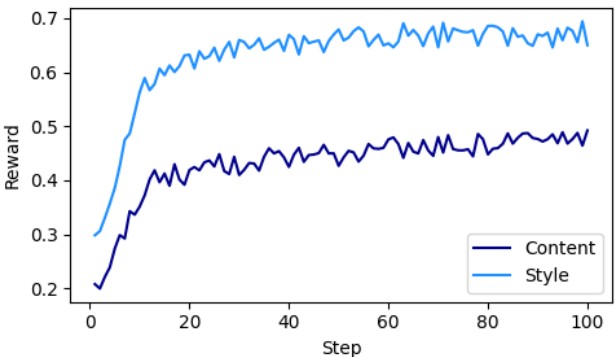

Figure 8: Content and style rewards of RLVRR during training, based on Qwen2.5-3B-Base.

---

[2]https://github.com/ollama/ollama

## F  CASE STUDY

In this case study, we analyze the performance of various methods, all based on the Qwen2.5-3B-Base model, using a sample instruction from AlpacaEval 2. Table 12 displays the responses generated by four different methods. The SFT model produces a concise and factually accurate answer, although it lacks detail and context regarding the name change. In contrast, models further trained with BLEU and RM yield incorrect responses, asserting that Facebook Corporation did not change its legal name and providing an inaccurate account of the rebranding process. Our proposed method, RLVRR, demonstrates a notable improvement by providing a response that is both factually accurate and comprehensive. Additionally, the response generated by our method is significantly shorter than those produced by BLEU and RM. This combination of detail, accuracy, and brevity highlights the superiority of our approach in delivering informative and precise answers.

Table 12: Generated responses from different methods for a sampled instruction in AlpacaEval 2.

| Instruction | Did Facebook corporation change its name? |
|---|---|
| **SFT** | Yes, Facebook Corporation changed its name to Meta Platforms, Inc. In 2021. |
| **BLEU** | No, the Facebook Corporation did not alter its legal name. Despite the fact that the company has introduced a diverse array of products and services beyond its primary social networking platform, which includes popular applications such as Instagram, WhatsApp, Messenger, and various others, the official name of the corporation continues to be Facebook, Inc. Nevertheless, in October 2021, Facebook made an announcement indicating their intention to modify their corporate brand name to better reflect their expanding range of offerings. As of now, the new brand name has not been disclosed to the public, and it is anticipated that the complete transition to this new name will require a significant amount of time. |
| **RM** | No, Facebook Corporation did not change its name. Facebook Corporation, also known as Facebook, Inc., is the parent company of the Facebook social media platform, and it has not gone through any significant name change. The company holds other online platforms and services such as Instagram and WhatsApp, but the name Facebook Corporation remains the same. There have been some changes in the branding, such as the rebranding of WhatsApp and Facebook Messenger with the Meta logo, but the parent company's name has not changed. The announcement made by Facebook in October 2021 about rebranding its parent company as Meta did not materialize, and Facebook Corporation still holds the name Facebook Corporation. |
| **RLVRR** | Yes, Facebook Corporation changed its name to Meta Platforms, Inc. In October 2021. The rebranding aims to reflect the company's expanded focus beyond social media and into building the "metaverse," a virtual space where people can interact and communicate in immersive, 3D environments. The new name "Meta" represents the company's ambition to drive breakthroughs in areas such as virtual and augmented reality. |

## G  LLM USAGE

We utilized large language models to support both manuscript polishing and data construction. In particular, the GPT-4o-mini API is employed to assist with the construction of the training dataset. Further details of this process are provided in Section 3 and Appendix A.2.

