# OpenReview forum: "From Verifiable Dot to Reward Chain: Harnessing Verifiable Reference-based Rewards for Reinforcement Learning of Open-ended Generation"
_ICLR.cc/2026/Conference — ICLR 2026 Poster_

### Official Review · Reviewer_kHas · 2025-10-31

**Soundness:** 3
**Presentation:** 3
**Contribution:** 2
**Rating:** 6
**Confidence:** 3

**Summary:**

This paper presents RLVRR, an approach to extend verifiable rewards to open domain generation by generating reward chains from high-quality references. The approach generates two kinds of rewards - style-based and content-based rewards. Content-based rewards are computed by extracting keywords from the references using an LLM and computing LCS with the response being evaluated, and the style-based rewards are generated as programmatic tests. Experiments on 10 benchmarks with Qwen and Llama models show that this method outperforms SFT-trained with 10 times more data, training with advanced reward models, and unifies the training of structured reasoning and open-ended generation, and generalizes more effectively while preserving output diversity.

**Strengths:**

1. Superior to SFT and alternative reward strategies. Also improves over DPO
2. Robustness across scales, model families, and initializations
3. Generalization to diverse tasks, including open-ended generation and knowledge-intensive reasoning, math, and coding tasks.
4. RLVRR doesn't compromise output diversity despite its reliance on verifiable references.
5. Low financial cost for data construction and low computational overhead for training (barely more than a random reward baseline).

**Weaknesses:**

1. Content-based rewards still rely on the mere notion of LCS with keywords, which seems gameable and doesn't capture deeper semantics. While the authors perform ablations that only calculate the fraction of keywords, which are completely vulnerable to reward hacking by models via keyword stuffing, LCS is not as easily hacked. However, what if cases where some sentences are permuted compared to the references, which leads to smaller sequential overlaps, get passed over even if they are virtually equivalent? Doing some reward value analysis with a few curated examples like that could be interesting.
2. Style-based rewards seem to be focusing on very surface-level characteristics. For example, in Figure 1, one of the stylistic properties evaluated is the markdown formatting, specifically the hierarchy of the title (exactly "###"). Could these rewards be overly strict? It doesn't make sense why "#" would be better than "###" given the question. Moreover, the program just uses "###" in answer, but what if the answer was something like "Decentralization doesn't f###ing work" (using ### for censoring rather than formatting).
3. Additional experiments to validate the importance and impact of the LLM-based weighting scheme. Currently, the comparison is only done between no-weighting and weighting, and even there, the differences **don't look statistically significant** except for FollowBench. It would be interesting to compare it with a reasonable random weighting scheme to see if the weighting is much better than random.
4. Computing "false positives" and "false negatives" of the reward chain: Doing a study with a small subset of data, where references are specifically engineered to break the reward functions (like the example given in weakness 2) and then seeing if the reward chain catches them or gives a false positive as well as references which are equivalent by the content is reorded (e.g sentnece permutation mentioned in weakness 1) and seeing if the reward chain misses them (false negatives) would be interesting. While the empirical results suggest that such scenarios might be unlikely in organic rollouts, it would be interesting to study whether this method has potential weaknesses that could surface in other scenarios or could be exploited for some sort of data poisoning attack.
5. The ablations show that keyword quality matters more than quantity. This suggests that the reference quality also indirectly matters (since bad references would yield bad keywords). This makes me wonder if these techniques would be useful at all for learning from noisy references. It feels like the style-based rewards would also overfit to the formatting style of noisy references, and the inferior keywords would quickly degrade performance.

**Questions:**

1. How did you arrive at the range of 3 to 6 functions for the style-based reward for the prompt shown in Figure 6?
2. Did you do any ablations where you gave example style-based reward functions to the LLM to give more inspiration on what type of functions to generate, and would that help prevent the generation of superfluous/unnecessary functions?

---

> ### Author Response · Authors · 2025-11-25
> **Response to Reviewer kHas (1/2)**
>
> We sincerely thank the reviewer for the positive assessment of our work as "superior, robust, and computational and cost-efficient" and for the thoughtful, constructive comments. We have carefully considered all points and have revised the manuscript accordingly.
>
> ---
>
> > W1: Content-based rewards still rely on the mere notion of LCS with keywords, which seems gameable and doesn't capture deeper semantics. What if cases where some sentences are permuted compared to the references, which leads to smaller sequential overlaps, get passed over even if they are virtually equivalent?
>
> We thank the reviewer for this excellent point regarding the robustness of content-based rewards against semantic-preserving variations like sentence permutation. Below, we clarify how our method addresses these concerns and provide additional experimental evidence to demonstrate its resilience.
>
> Our content reward is designed to be invariant to syntactic or structural variations like sentence reordering:
>
> - **Key Point Extraction and Keyword Alignment:​** Instead of relying solely on sequential keyword matching, our method first decomposes the question into core "key points" and extracts a set of representative keywords/concepts for each point. The reward computation then uses the Longest Common Subsequence (LCS) at the **key point level** to evaluate content alignment. This ensures that permutations (e.g., swapping the order of key points) do not penalize semantically equivalent responses. For example, the following semantically equivalent but structurally different responses received identical content rewards:
>   ```
>   Question: What are the advantages of transformer-based models in terms of capturing long-range dependencies and parallelized training?
>   Response 1: Transformer-based models excel at capturing long-range dependencies primarily due to their core self-attention mechanism... Regarding parallelized training, Transformer models hold a significant advantage over their sequential counterparts...
>   Response 2: Regarding parallelized training, Transformer models hold a significant advantage over their sequential counterparts... Transformer-based models excel at capturing long-range dependencies primarily due to their core self-attention mechanism...
>   ```
>
> - **Multiple Reference Support:​** As detailed in Section 3.2 (Equation 4), our reward mechanism incorporates multiple reference answers. This explicitly accounts for diversity in legitimate phrasing and structure, preventing the model from overfitting to a single expression format.
>
>
> > W2: Style-based rewards seem to be focusing on very surface-level characteristics.
>
> We appreciate the reviewer's concern. The example in Figure 1 may unintentionally give the impression that RLVRR enforces literal string patterns like "###". In practice, this is not how style rewards work.
>
> **(1) Style rewards evaluate structural properties, not raw tokens.**
> As described in Section 3.3, the verifier code is generated by an LLM and checks style/formatting requirements (e.g., "has section titles") rather than exact substrings. Thus, a case like "f###ing" would not trigger a formatting penalty, because the verifier operates on structure, not literal matches.
>
> **(2) Style constraints are intentionally lightweight and task-adaptive.**
> The verifier is synthesized from the reference response, so formatting rules only appear in tasks where structure is actually relevant. When the reference does not use headings, no such rule is generated.
>
> **(3) Empirically, RLVRR does not overfit stylistic patterns.**
> As shown in Table 7, RLVRR maintains Self-BLEU comparable to RM and the _Instruct_ model, confirming that our style reward does not reduce output diversity or enforce brittle formatting.
>
> We will clarify in the revision that Figure 1 is a simplified illustration and does not reflect literal pattern matching.
>
> > W3: Additional experiments to validate the importance and impact of the LLM-based weighting scheme.
>
> To directly investigate its importance, we have supplemented our ablation studies with a new experiment comparing it against a random weighting scheme. The results show that the random weighting scheme underperforms our proposed LLM-based approach. In fact, its performance is comparable to having no weighting scheme at all ("no weight"). This provides strong evidence that the intelligent, learning-informed weighting provided by the LLM is a key contributor to the overall performance of RLVRR.
>
> | Method | AlpacaEval 2 | Arena-Hard | MT-Bench | IF-Eval | FollowBench | Open-ended Avg. |
> |--------|-------------|-------------|-------------|-------------|-------------|-------------|
> | RLVRR (Ours) | **23.7** | **35.3** | **7.6** | **37.7** | **51.2** | **31.1** |
> | -w/o style reward | 19.6 | 28.5 | 6.6 | 36.6 | 50.1 | 28.3 |
> | -w/o weight in style | 22.2 | 35.0 | 7.6 | 36.2 | 48.6 | 29.9 |
> | **-w random weight in style (New)** | 22.1 | 34.8 | 7.5 | 36.0 | 48.1 | 29.7 |

---

> > ### Author Response · Authors · 2025-11-25
> > **Response to Reviewer kHas (2/2)**
> >
> > > W4: "false positives" and "false negatives" in reward chains.
> >
> > We would like to emphasize that RLVRR significantly mitigates the risks of false positives through its design. Specifically, the content reward is calculated by **first extracting key points and then identifying the corresponding keywords for each key point**. The reward computation then uses the Longest Common Subsequence (LCS) at the **key point level** to evaluate content alignment. This process ensures that small variations, such as sentence or paragraph reordering, do not negatively impact the reward calculation. As a result, even when content is rearranged, the reward chain can still accurately capture the content fidelity, preventing issues like false negatives.
> >
> > Additionally, **RLVRR leverages multiple references in the reward computation**, which further enhances its robustness. By comparing the generated response to several reference responses, RLVRR ensures that small variations in phrasing or structure (e.g., sentence permutation) do not lead to inaccurate reward assignments. This multiple-reference approach helps to account for linguistic diversity while maintaining the integrity of the content.
> >
> > While we believe that false positives and false negatives are unlikely to arise in typical scenarios, we agree that testing the system with deliberately engineered cases, such as data poisoning attacks or adversarial examples, would be a useful direction for future work. This would allow us to further evaluate the robustness of the RLVRR framework against potential exploits or weaknesses in adversarial settings.
> >
> > > W5: Would RLVRR be useful at all for learning from noisy references?
> >
> > We appreciate the reviewer's insightful concern regarding noisy references. Indeed, RLVRR benefits from high-quality references, as keyword precision influences content reward quality. However, several design choices of RLVRR substantially **mitigate sensitivity to noisy data**, making the framework more robust than it may appear:
> >
> > (1) **Robustness via structured, verifiable rewards rather than verbatim imitation.**
> > Unlike SFT—which directly imitates noisy sequences—RLVRR only extracts _verifiable semantic units_ (keywords) and _style criteria_ from references. These signals are rule-based and coarse-grained, which makes them inherently more noise-tolerant. Even when keywords are imperfect, the LCS-based content reward evaluates relative semantic overlap, avoiding overfitting to exact wording or formatting noise.
> >
> > (2) **Multiple references significantly dampen noise effects.**
> > By incorporating multiple references during training, RLVRR can reduce the impact of noisy or suboptimal references. This approach ensures that a single noisy reference does not unduly influence the overall reward signal. Our experiments show that even with a limited number of references, RLVRR outperforms methods like RM, demonstrating that the system can tolerate and generalize well in the presence of variability in references.
> >
> > (3) **Empirically, RLVRR remains strong even when reference quality drops.**
> > As noted in our ablation (Table 3), even when replacing GPT-4o-mini references with the weaker Llama3-70B-Instruct references, RLVRR still achieves substantial gains over SFT and RM-based methods. The performance decreases modestly (31.1 → 28.9), indicating graceful degradation rather than failure.
> >
> > In future work, we plan to further refine these components and investigate how noisy data could be handled more effectively without compromising performance.
> >
> > > Q1: How did you arrive at the range of 3 to 6 functions for the style-based reward for the prompt shown in Figure 6?
> >
> > The range of 3 to 6 stylistic functions was determined empirically through extensive experimentation, with the goal of balancing complexity and generalization. We found that having 3 to 6 functions allows the model to learn meaningful stylistic properties (e.g., length constraints, formatting, structure) without overfitting to any single stylistic trait.
> >
> > > Q2: Did you do any ablations where you gave example style-based reward functions to the LLM to give more inspiration on what type of functions to generate, and would that help prevent the generation of superfluous/unnecessary functions?
> >
> > We did explore the idea of providing the LLM with example style-based reward functions to guide the generation of stylistic checks. However, we found that allowing the LLM to generate these functions autonomously resulted in better generalization and diversity of stylistic evaluations.
> >
> > ---
> >
> > We hope this clarifies the points raised and demonstrates that the RLVRR framework is robust and effective, with strategies in place to address the concerns highlighted by the reviewer. We are grateful for the reviewer’s constructive feedback and will continue to refine RLVRR in future iterations based on these valuable insights.

---

### Official Review · Reviewer_SpgL · 2025-11-01

**Soundness:** 3
**Presentation:** 3
**Contribution:** 3
**Rating:** 6
**Confidence:** 3

**Summary:**

The paper proposes RLVRR, reinforcement learning with verifiable reference-based rewards, to bring RL with verifiable signals (RLVR) beyond math/code into open-ended generation. Instead of a single “verifiable dot,” RLVRR builds a reward chain from a high-quality reference: a content reward (reference-derived keywords/points checked via rule-based matching) plus a style reward (small, LLM-generated, verifiable Python checks like length/format), aggregated for GRPO training with a KL penalty to a reference model. This yields stable, cheap, and hack-resistant supervision and shows gains across >10 benchmarks with Qwen and Llama models.

**Strengths:**

- Splits reward into content (reference-derived key points/keywords) and style (verifiable code checks), avoiding fuzzy learned RMs during training.
- On open-ended and other tasks, RLVRR improves over SFT (even with 10× more data) and over other baselines, and computation overhead is very small
- Mixing math RLVR data with open-ended RLVRR produces competitive math and open-ended performance, showing the framework extends the RLVR paradigm.
- Quite cheap data contruction cost (and even open-source LLMs can replace GPT-4o-mini) to build verifiable components with comparable outcomes.

**Weaknesses:**

- The content reward relies on information extracted by the reference LLM; performance may hinge on that accuracy/quality. You show a random baseline, but I’m curious how small perturbations/noise in the extracted cues affect results.
- To cut compute, the reward checks text form rather than semantics. If the policy improves and paraphrases with different wording, this approach may hit a ceiling by penalizing valid semantic matches that use different tokens.
- What happens if the policy is an instruction-tuned model instead of a base model? I wonder how that changes reward alignment and behavior, especially for formatting/style constraints.

**Questions:**

see above

---

> ### Author Response · Authors · 2025-11-25
> **Response to Reviewer SpgL**
>
> Thank you for your thoughtful and constructive comments! We appreciate your positive feedback on the effectiveness and efficiency of our work and have carefully addressed your specific points below.
>
> ---
>
> > W1: The content reward relies on information extracted by the reference LLM; performance may hinge on that accuracy/quality. You show a random baseline, but I’m curious how small perturbations/noise in the extracted cues affect results.
>
> To thoroughly investigate this, we supplemented the random baseline experiment with a more granular noise-injection analysis. We systematically replaced extracted keywords with randomly sampled non-keywords at different perturbation rates (10%, 20%, 40%). The results demonstrate that our method maintains promising performance even with a 20% perturbation level, confirming its robustness to moderate noise.
>
> | Method                                       | Open-ended Avg. |
> |----------------------------------------------|-----------------|
> | RLVRR | **31.1** |
> | RLVRR (replace 10% keywords) | 30.5 |
> | RLVRR (replace 20% keywords) | 29.4 |
> | RLVRR (replace 40% keywords) | 27.3 |
> | random extraction baseline | 27.0 |
>
> > W2: To cut compute, the reward checks text form rather than semantics. If the policy improves and paraphrases with different wording, this approach may hit a ceiling by penalizing valid semantic matches that use different tokens.
>
> This is an excellent question regarding the semantic flexibility of our reward. Our mechanism is designed to focus on core content elements (keywords/entities), which are more stable across paraphrasing. To quantitatively validate this, we conducted a new experiment measuring the Jaccard similarity of keywords extracted from answers generated by three different LLMs (GPT-4o-mini, Qwen3-235B, DeepSeek-V3) on 100 training examples. The average keyword similarity was **0.56**, significantly higher than the full-text similarity (**0.23**), confirming that core content remains consistent even with different phrasing.
>
> Furthermore, **we have extended our content reward to leverage multiple references**​ (Section 3.2, Eq. 4), which explicitly improves tolerance to phrasing variations. The ablation study (Table 4) shows that performance remains robust (30.7 vs. 31.1) even with a single reference, demonstrating the method's ability to handle semantic matches effectively without being overly rigid.
>
> > W3: What happens if the policy is an instruction-tuned model instead of a base model? I wonder how that changes reward alignment and behavior, especially for formatting/style constraints.
>
> We have conducted experiments based on instruction-tuned models, including Qwen2.5-3B/7B-Instruct in Table 1 and Llama3.1-8B-Instruct in Table 9. The results show that RLVRR consistently improves performance on both open-ended benchmarks and downstream tasks, including instruction-following metrics like IFEval and FollowBench. This confirms that our approach effectively further refines pre-existing instruction-following and formatting capabilities, though the absolute gains are somewhat smaller compared to base model initialization.
>
> ---
>
> We hope these clarifications address your concerns and enhance your understanding of our approach. Thank you again for your valuable feedback!

---

### Official Review · Reviewer_YJen · 2025-11-02

**Soundness:** 3
**Presentation:** 3
**Contribution:** 3
**Rating:** 6
**Confidence:** 3

**Summary:**

This paper propose a novel RLVRR framework that extend RLVR to domains where reference answers are open-ended and long-form. RLVRR includes two kinds of rewards: content reward that maximize LCS between ordered keywords mentions in reference answer and generated response, style reward that constrains factors like formats and length based on the reference answer style. Comprehensive experiments demonstrate that RLVRR surpass BLUE, RM, GRM, RLPR baselines.

**Strengths:**

1. The proposed method is more effective compared with competitive baselines like RM and GRM, while also efficient.
2. Experiments on both base models and instruct models show good performance of RLVRR.

**Weaknesses:**

1. Lacking the reward quality analysis. For example, how RLVRR achieves better results than the GRM setting that also rely on GPT-4o-mini, is it because the reward quality of RLVRR is better?
2. It requires proprietary APIs to generate key points information, which might also hinder its scalability.
3. Is RLVRR more computation efficient compared with the SFT baseline, regarding the training costs. Since SFT is much faster compared with RL which requires slow online response generation.

**Questions:**

1. How does RLVRR handle the flexible natural of languages, as the model could state the same thing in different word choices compared with the reference answer. In this case, the LCS method fails to capture the semantically equivalent  keyword mentions.

---

> ### Author Response · Authors · 2025-11-25
> **Response to Reviewer YJen**
>
> Thank you for your valuable comments! We're glad that you found our work effective and efficient. We have listed our response to your concerns as follows.
>
> ---
>
> > W1: Lacking the reward quality analysis. For example, how RLVRR achieves better results than the GRM setting that also relies on GPT-4o-mini, is it because the reward quality of RLVRR is better?
>
> We appreciate your comment on the reward quality analysis. In Section 5.3 of our paper, we perform an analysis of the reward and response length curves during RL training with different methods. Our results demonstrate that RLVRR achieves a more stable and significant increase in reward compared to other methods, highlighting its superior ability to provide consistent and high-quality learning signals.
>
> To address your specific question about the comparison to GRM, we further analyze the dynamics of GRM during training. We observe that GRM exhibits a trend similar to RM, where both the reward and the response length steadily increase, indicating that GRM tends to favor longer responses—likely due to over-reliance on superficial heuristics rather than truly improving content quality. In contrast, RLVRR avoids this issue, as its reward structure encourages semantic content alignment without incentivizing verbosity. We will add these details to the final version of the paper to provide a clearer comparison between RLVRR and GRM.
>
> > W2: It requires proprietary APIs to generate key points information, which might also hinder its scalability.
>
> As demonstrated in Table 4, RLVRR is robust across different LLMs, including both proprietary and open-source models. Specifically, when we substitute GPT-4o-mini with **Llama3-70B-Instruct**, a powerful open-source alternative, the performance of RLVRR remains competitive, showing that our approach does not depend on proprietary APIs for high-quality results. This reinforces the scalability of RLVRR, as it can be effectively applied even in settings with limited access to proprietary models.
>
> > W3: Is RLVRR more computation efficient compared with the SFT baseline, regarding the training costs. Since SFT is much faster compared with RL which requires slow online response generation.
>
> You are correct that SFT training is faster than RL due to the online nature of RL training. In our experiments, we find that the time taken for RL training on a 10k dataset is comparable to the time taken for SFT training on a 100k dataset. Despite the higher computational cost of RL, our proposed RVLRR still outperforms SFT trained with 10$\times$ more data and achieves much better in generalization to diverse tasks, as shown in Table 1. Moreover, the efficient training dynamics of RLVRR, especially in terms of minimizing the reliance on reward models, make it a more scalable solution in the long run.
>
> > Q1: How does RLVRR handle the flexible natural of languages, as the model could state the same thing in different word choices compared with the reference answer. In this case, the LCS method fails to capture the semantically equivalent keyword mentions.
>
> This is an excellent question. Our content reward mechanism in RLVRR is designed to handle such language flexibility. It measures the inclusion and order of key content elements (keywords or entities) that are extracted from references, using a two-level hierarchical keyword extraction approach. These core concepts are relatively deterministic, ensuring that even when the phrasing changes, the critical elements of the content are captured.
>
> To further validate this, we conducted an experiment where we sampled 100 examples from the training data and obtained answers from three different LLMs: GPT-4o-mini, Qwen3-235B-A22B-Instruct-2507, and DeepSeek-V3.1. We extracted keywords from these answers using GPT-4o-mini. The results show an average Jaccard similarity of **0.56** between the keyword sets of each pair of LLMs, compared to a much lower similarity of **0.23** when considering the entire answer. This indicates that while word choices may differ, the core semantic content remains consistent across models.
>
> Additionally, to improve robustness and handle variations in word phrasing, **we extend our content reward mechanism from a single reference to multiple references**, as described in Section 3.2 and formalized in Equation (4). This extension ensures that RLVRR is tolerant to variations in reference phrasing while maintaining strict content fidelity. Our ablation study in Table 4 further confirms this, showing that even when using a single reference for content reward computation, the performance remains robust, with only a marginal drop from 31.1 to 30.7. This demonstrates RLVRR’s resilience to reference variability and its ability to handle flexible language effectively.
>
> ---
>
> We hope these clarifications address your concerns and enhance your understanding of our approach. Thank you again for your valuable feedback!

---

### Official Review · Reviewer_ExFE · 2025-11-04

**Soundness:** 3
**Presentation:** 4
**Contribution:** 3
**Rating:** 8
**Confidence:** 4

**Summary:**

The paper introduces RLVRR (Reinforcement Learning with Verifiable Reference-based Rewards), an extension of RLVR (Reinforcement Learning with Verifiable Rewards). While RLVR works well for reasoning tasks (like math and code) where correctness can be verified from the final result ("verifiable dots"), it struggles with open-ended generation where there is no single ground truth. RLVRR tackles this challenge by introducing reference-based verifiable reward chains. The core idea is that instead of rewarding only the final verifiable answer, RLVRR derives an ordered linguistic reward chain from reference responses.

The method decomposes the reward into two verifiable components: (a) content reward that measures inclusion and order of key content elements (keywords or entities) extracted from references, using a two-level hierarchical keyword extraction (key points --> keywords). Alignment is computed via Longest Common Subsequence (LCS) between reference and generated keywords; (b) style reward that checks adherence to stylistic properties using small, verifiable Python functions automatically generated by an LLM (e.g., enforcing markdown format, word length, structure). These are weighted based on LLM-estimated importance. The policy is trained with the GRPO algorithm, optimizing expected rewards with a KL regularization term relative to a reference model.

The authors evaluate their approach using Qwen2.5 (3B, 7B) and Llama3.1 models. They use 10K RL training samples constructed from a 100K instruction-following corpus. The baselines compared are SFT (10K & 100K), Random, BLEU-based, Reward Model (RM), Generative Reward Model (GRM), RLPR, and DPO. The benchmarks used are focused on open-ended generation: AlpacaEval 2, Arena-Hard, MT-Bench, IFEval, FollowBench, and other domains: MMLU (knowledge), ARC (reasoning), MATH (math), HumanEval (code). Overall, RLVRR outperforms all baselines, including SFT (even with 10× more data) and RLHF-based methods.

**Strengths:**

- The paper is very well-written and easy to understand. In particular, Section 3, the pipeline diagram, and the prompts provided in the Appendix make it easy for readers to follow and grasp the details.

- RLVRR’s "reward chain" bridges reasoning-style verifiability with open-ended text generation, which is a key step beyond single-dot verification. It removes dependency on trained reward models, reducing cost, reward hacking, and brittleness.

- RLVRR outperforms SFT, DPO, RLHF-style, and BLEU-based methods across 10+ benchmarks, with minimal compute overhead. It also demonstrates cross-domain robustness, integrates with math reasoning (RLVR), and maintains generation diversity.

**Weaknesses:**

- One major weakness is that RLVRR captures only rule-based content and style fidelity; it may miss deeper semantic or ethical nuances that require human judgment. For example, the "content reward" relies entirely on an LLM judge, which is responsible for extracting critical keywords from the reference responses (the paper claims these keywords capture the "semantics" of the reference). However, this does not truly capture semantics for reasoning tasks like mathematical reasoning (GSM8K, MATH, OlympiadBench); it may only work for tasks like essay generation or plain natural language responses. For mathematical reasoning, adhering to certain tokens does not in any way ensure the generated response is correct, and there are no inherent discrepancies. The idea of extracting key points and then keywords is a reasonable abstraction, but it is unclear why it should improve performance on mathematical reasoning tasks. Furthermore, for these tasks, shouldn't the "verifiable dots" also be included in the reward? The final answer is at least as important as, if not more important than, adhering to key points or keywords.

- For the style reward, the authors use an LLM to generate a Python verifier function that checks properties like answer length and markdown formatting. These rewards capture surface-level stylistic/formatting patterns rather than the semantic equivalence of the generated and reference responses. It is not clear how these align with tasks like Knowledge, Reasoning, or Math benchmarks. While such patterns may be useful for open-ended instruction-following tasks, they make little sense for Knowledge, Reasoning, or Math tasks.

- The metrics used for the different benchmarks should be clearly defined, and more appropriate/fine-grained metrics for instruction-following should be considered. For example, "length-controlled win rate" or using "GPT-4.1-mini as the evaluation judge" are weak metrics for evaluating instruction following. They do not effectively capture how well the instruction is followed or, more importantly, whether the "content" and "style" of the generated responses are adequate.

**Questions:**

See weaknesses.

---

> ### Author Response · Authors · 2025-11-25
> **Response to Reviewer ExFE**
>
> Thank you for your insightful and constructive comments! We're glad that you found our work well-written, effective, and efficient. We have tried our best to elaborate on the unclear points and revised our paper accordingly.
>
> ---
>
> > W1 & W2: The content and style rewards of RLVRR may only work for tasks like essay generation or plain natural language responses rather than reasoning tasks like math.
>
> We would like to emphasize that **RLVRR is designed specifically for open-ended generation tasks**, where traditional RLVR methods face challenges due to the lack of unambiguous ground truth. In contrast, reasoning tasks such as mathematical problem solving (e.g., GSM8K, MATH, OlympiadBench) typically involve clear, verifiable answers, making RLVR more suitable for these domains. The RLVRR framework extends reinforcement learning with verifiable rewards to open-ended generation, where no single "verifiable dot" exists, and answers cannot be verified by a simple correctness check.
>
> In reasoning tasks, RLVR's use of a verifiable final result is highly effective, as the correctness of the final answer directly guides learning. RLVRR, however, introduces an alternative approach that captures the _content_ (core concepts or keywords) and _style_ (formatting and linguistic structure) of a response, which is more appropriate for open-ended generation tasks.
>
> We also conducted additional experiments with RLVRR applied to 10K math data, and the results show that while RLVRR improves performance on math tasks from 46.1 to 51.1, it is still outperformed by RLVR (which checks the final answer). However, RLVRR dramatically outperforms RLVR on open-ended tasks. Additionally, when we combine RLVR for reasoning tasks and RLVRR for open-ended tasks (unified training), RLVRR even surpasses _Qwen2.5-3B-Instruct_ that was trained on millions of samples. This demonstrates that **RLVRR effectively integrates with RLVR for unified training across both structured reasoning and open-ended generation tasks**.
>
> Here's a breakdown of the results:
> | Method                                       | Math Avg.  | Open-ended Avg. |
> |----------------------------------------------|-------|-----------------|
> | _Qwen2.5-3B-Base_                            | 46.1  | 9.6             |
> | $\hookrightarrow$ GRPO - 10K math (RLVR)                       | 51.9  | 22.6            |
> | $\hookrightarrow$ GRPO - 10K math (RLVRR)                       | 51.1  | 25.3            |
> | $\hookrightarrow$ GRPO - 10K open-ended (RLVRR)                | 49.8  | **31.1**            |
> | $\hookrightarrow$ GRPO - 5k math (RLVR) + 5k open-ended (RLVRR) | **51.9** | 30.7       |
> | $\hookrightarrow$ GRPO - 5k math (RLVR) + 5k open-ended (RM)   | 50.4  | 28.2            |
> | _Qwen2.5-3B-Instruct_                      | 51.4  | 29.3            |
>
> These results validate that RLVRR can be seamlessly integrated with RLVR, showing improvements on open-ended tasks while maintaining strong performance on reasoning tasks. We will incorporate this clarification into the revised submission to enhance understanding.
>
>
> > W3: The metrics used for the different benchmarks should be clearly defined, and more appropriate/fine-grained metrics for instruction-following should be considered.
>
> The evaluation metrics for different benchmarks are defined in the main text (lines 224–227), with extended details provided in Appendix B. We believe these sections fully describe our metric choices.
> To evaluate instruction-following capabilities, we utilize two widely adopted benchmarks: **IFEval** [1] and **FollowBench** [2].
> - **IFEval** is specifically designed to assess instruction-following LLMs across a variety of instructions that involve different lexical and formatting constraints. It provides a **rule-based metric** for evaluating how well models handle instructions that demand specific content and formatting.
> - **FollowBench** is a more granular benchmark that evaluates models' ability to follow constraints across five types and five difficulty levels. This fine-grained evaluation allows for a deeper understanding of the model's performance, particularly in adhering to specific instruction nuances.
>
> We report results for both benchmarks in **Table 1**, demonstrating that RLVRR significantly improves instruction-following performance over other baselines. Specifically, RLVRR enhances the ability of the model to adhere to diverse instruction constraints, leading to substantial gains in both IFEval and FollowBench metrics. We will provide further clarification in the revised manuscript regarding the metrics and their relevance to our tasks.
>
> ### References:
> [1] Instruction-Following Evaluation for Large Language Models. Zhou et al. arXiv 2023.
>
> [2] FollowBench: A Multi-level Fine-grained Constraints Following Benchmark for Large Language Models. Jiang et al. ACL 2024.

---

### Meta-Review · Area_Chair_vYvT · 2026-01-07

**Summary:**

This paper introduces a generalisation of RLVR wherein answers are compared against multiple references, semantically processed by a LLM-derived evaluator, to compute the verification reward. The main points made by the reviewers are the following.

Strengths:
1. Good writing (ExFE)
2. Interesting/useful idea (ExFE, SpgL, kHas)
3. Solid experiment results (ExFE, YJen, SpgL, kHas)

Weaknesses/concerns:
1. Rule-based rewards can miss some semantic aspects, questions on semantic equivalence (ExFE, YJen, SpgL, kHas)
2. Both content and style rewards not applicable to formal reasoning settings, or even to mathematical reasoning in language (ExFE)
3. Questions about evaluation metrics (ExFE)
4. Analysis of the reward quality, questions on sensitivity to reference LM and reference set (YJen, SpgL, kHas)
5. Scalability and computational efficiency questions (YJen)
6. Significance of results on weighting scheme (kHas)

The authors responded to all of these concerns, as detailed below, and think that the reviewers -- already all recommending acceptance -- would have been satisfied with the responses. I therefore recommend acceptance, but would like to see some of the improvements made below in the final version.

**Reviewer Concerns:**

1. This, while an open-ended question, was answered partially by some interesting discussion in the rebuttal, which shows that the semantic processing (keyword extraction) and use of multiple references mitigate the sensitivity to surface form variations.
2. Addressed: the authors say such tasks are not in the scope considered, yet the method is still applicable to mathematical reasoning tasks (new experiment).
3. Addressed in rebuttal: the evaluation metrics are standard.
4. This is partially answered and some interesting discussion is given, but the authors may still want to consider doing the main suite of experiments with an open-source reference model for better reproducibility and understanding of the method.
5. Answered in rebuttal: as expected, the RL-based method is more expensive than SFT, but the authors argue that it is still amortisedly beneficial.
6. Addressed by a new experiment, but the issue remains of none of the experiments in the paper coming with statistical significance analysis (error intervals).

Other minor questions also seem satisfactorily answered.

**Reviewer Scores:**

The original scores were 8 (ExFE), 6, (YJen), 6 (SpgL), 6 (kHas). None of the reviewers responded to the rebuttals. I think it is possible that all three of the lower scores would have been increased to 8, particularly kHas, who had the greatest volume of concerns resolved by the responses.

---

### Decision · Program_Chairs · 2026-01-26

Accept (Poster)